# Cloning and Characterization of Fructose-1,6-Bisphosphate Aldolase from *Euphausia superba*

**DOI:** 10.3390/ijms231810478

**Published:** 2022-09-09

**Authors:** Jikun Xia, Wanmeng Xin, Fang Wang, Wancui Xie, Yi Liu, Jiakun Xu

**Affiliations:** 1College of Marine Science and Biological Engineering, Qingdao University of Science and Technology, Qingdao 266042, China; 2Key Lab of Sustainable Development of Polar Fisheries, Ministry of Agriculture and Rural Affairs, Yellow Sea Fisheries Research Institute, Chinese Academy of Fishery Sciences, Lab for Marine Drugs and Byproducts of Pilot National Lab for Marine Science and Technology, Qingdao 266071, China; 3State Key Laboratory of Biocatalysts and Enzyme Engineering, School of Life Sciences, Hubei University, Wuhan 430062, China

**Keywords:** fructose-1,6-bisphosphate aldolase, *Euphausia superba*, heterologous expression, molecular simulation, point mutation

## Abstract

Fructose-1,6-bisphosphate aldolase (EC 4.1.2.13) is a highly conserved enzyme that is involved in glycolysis and gluconeogenesis. In this study, we cloned the fructose-1,6-bisphosphate aldolase gene from *Euphausia superba* (EsFBA). The full-length cDNA sequence of EsFBA is 1098 bp long and encodes a 365-amino-acid protein. The fructose-1,6-bisphosphate aldolase gene was expressed in *Escherichia coli* (*E. coli*). A highly purified protein was obtained using HisTrap HP affinity chromatography and size-exclusion chromatography. The predicted three-dimensional structure of EsFBA showed a 65.66% homology with human aldolase, whereas it had the highest homology (84.38%) with the FBA of *Penaeus vannamei*. Recombinant EsFBA had the highest activity at 45 °C and pH 7.0 in phosphate buffer. By examining the activity of metal ions and EDTA, we found that the effect of metal ions and EDTA on EsFBA’s enzyme activity was not significant, while the presence of borohydride severely reduced the enzymatic activity; thus, EsFBA was confirmed to be a class I aldolase. Furthermore, targeted mutations at positions 34, 147, 188, and 230 confirmed that they are key amino acid residues for EsFBA.

## 1. Introduction

Fructose-1,6-bisphosphate aldolase (FBA, EC4.1.2.13) is a key enzyme in energy metabolism and is ubiquitously found. Glycolysis is a universal and fundamental biochemical pathway in carbohydrate metabolism. Fructose-1,6-bisphosphate aldolase (FBA) catalyzes the first step of glycolysis, which involves the cleavage of the six-carbon sugar backbone (fructose-1,6-bisphosphate) to generate two three-carbon intermediates [1,2,3,4]. Based on its catalytic mechanism and evolutionary origin, FBA can be divided into two types: class I aldolases and class II aldolases [5]. Class I FBAs are usually found in higher eukaryotes (animals and plants), where they use active-site lysine residues to stabilize the reaction intermediates through the formation of the Schiff base [6]. Its enzymatic activity can be inhibited by borohydride (usually NaBH_4_). Eukaryotic FBAs are homotetramers in which the subunits adopt the folding of parallel (βα)8-(TIM)-barrel [7,8,9]. Class II aldolases, found primarily in microorganisms as homodimers, require divalent metal cations (typically Zn^2+^) as cofactors to stabilize the carbonate intermediates formed by the substrate within the active site [10]. Thus, the activity of class II is competitively inhibited by the metal ion chelator EDTA [11]. Class I and class II aldolases have the same overall fold and catalyze the same overall reaction, but they do not share any significant sequence homology or common catalytic residues. Both types of aldolases can catalyze the cleavage of fructose-1,6-bisphosphate (FBP) to dihydroxyacetone phosphate (DHAP) and glyceraldehyde-3-phosphate (G3P) (Figure 1), which could provide ATP and substrates for the anabolism of biological substances [12,13,14]. In addition, a third class of aldolases has been reported, which are designated as class IA and are present in archaea (e.g., *Thermomycetes*, *Fireball*). These have a similar mechanism of action as class I aldolases and have been reported to exist as homo-octamers, decamers, or even higher oligomers [15,16].

As the most well-known DHAP-dependent aldolase [17], FBA has been used to produce important sugar derivatives, such as iminosugars, cyclitols, and complex natural products. Li [18] used FBA and three other enzymes to generate rare ketose molecules from various aldehyde substrates through a one-pot four-enzyme synthesis. FBA was shown to play an important role in combating hypoxia in sea turtles, and FBA increased the glycolytic output and alleviated hypoxia by increasing the level of FBA during hypoxia [19]. The upregulation of FBA gene expression in the muscle of *Penaeus prawn* infected with the white spot syndrome virus (WSSV) indicated that the FBA gene plays a role in the WSSV resistance of *Penaeus prawn* and is a disease-resistant gene; moreover, the physical interaction between aldolase and RNA polymerase II was thought to play a role in the regulation of hypoxic metabolism [20].

*Euphausia superba* is a small marine planktonic crustacean that lives in Antarctic waters and is known as the largest renewable animal protein pool for humans because of its large resources and high protein quality [21]. *E. superba* are extremely abundant and found throughout the Southern Ocean around Antarctica, with an annual biological catch of 100 million tons; therefore, it plays a key role in the Antarctic food web and is an important resource for fisheries [22,23]. *E. superba* contains many bioactive components, such as proteins, fatty acids, vitamins, minerals, and enzymes, and has become a hot topic of research in recent years [24,25,26]. *E. superba* can live in extreme environments and possesses qualities such as frost resistance and oxidation resistance [27,28,29]. In particular, the FBA from *E. superba* has great potential in the field of catalysis [30]. Because *E. superba* itself is prone to autoproteolysis, and it is difficult to isolate and extract the enzyme [31,32], scientists have attempted to develop *E. superba*-derived enzymes through genetic technology; FBA has not been studied in *E. superba*. Most studies on FBA have focused on heterologous expression in prokaryotes, mainly for vaccine development and diagnosis [33,34,35], whereas little research has been completed on eukaryotic FBA. In the past few years, scientists have been trying to achieve heterologous expression of key enzymes from *E. superba*, but no progress has been made. Therefore, studying FBA is not only conducive to understanding the evolutionary position of *E. superba* in the biosphere, but also provides a theoretical basis for the further application of fructose-1,6-bisphosphate aldolase. To our knowledge, this is the first report on the heterologous expression of FBA from *E. superba* (EsFBA).

## 2. Results and Discussion

### 2.1. Cloning and Bioinformatics Analysis

FBA in the EsFBA gene was validated by subcloning using krill cDNA as a template. The full-length EsFBA cDNA was 1098 bp long, containing 365 AA, with a molecular weight (Mw) of 39,820.37 Da. The theoretical pI was 7.59. The 365 AA include 43 positively charged residues (Asp + Glu) and 44 negatively charged residues (Arg + Lys). The instability index was 24.83 (<40), indicating that the EsFBA protein was stable. The aliphatic index was 81.56, indicating that the protein is soluble in lipids. We used ProtScale to perform a hydrophobic analysis of the protein and learned that it possesses more hydrophilic residues than hydrophobic residues, thus presuming that EsFBA is a hydrophilic protein. There was no *O*-glycosylation site; however, an *N*-glycosylation site was present at 73-NVSG. SWISS-MODEL was used to predict the 3D structure of the EsFBA (Figure 2A). Human A-type FBA was used as a template (Figure 2B). EsFBA is 65.38% homologous to the template, and the predicted structure of the EsFBA 3D model consists of 10 main β-stranded nuclei surrounded by 13 α-helices. The tetrameric structure is shown in Figure 2C,D.

The sequences obtained from sequencing were compared using NCBI Blast, and partial sequences with high similarity to FBA of *E. superba* were obtained. The highest similarity was found in *Penaeus vannamei*, followed by *Penaeus monodon*, zebrafish, *Pike tritomata*, and *Drosophila melanogaster*. The amino acid sequences encoded by the EsFBA were compared with each species for a multiple-sequence homology using DNAMAN8.0 (Figure 3).

The phylogenetic tree was made using MEGA-X for the above sequences (Appendix A). The results showed that the closest relatives to *Euphausia superba* were *Penaeus vannamei* with an 84.38% amino acid sequence similarity. It was 65.66% similar to Homo sapiens aldolase A, 59.34% similar to Homo sapiens aldolase B, and 65.38% similar to Homo sapiens aldolase C. From the phylogenetic tree and amino acid sequence similarity, it is clear that EsFBA belongs to the FBA family of genes, and the evolutionary relationship of this gene is generally consistent with the evolutionary status of the traditional species.

### 2.2. Protein Expression and Purification

The codon-optimized target gene fragment was 1131 bp in length, and the molecular weight (Mw) of the protein was 41,105.70. The synthesis of the optimized gene resulted in an amplification product (Appendix A). The recombinant plasmid pET28-sfGFP-EsFBA was constructed and expressed in *E. coli* BL21 (DE3), as described in the methods Section 3.4. After isopropyl β-D-1-thiogalactopyranoside (IPTG) induction, HisTrap HP purification, and SDS-PAGE, a single band with a relative molecular mass of 45 kDa (Figure 4) was obtained. Compared with the predicted protein molecular weight of 41 kDa, this slight increase in molecular weight is due to the expression tag.

### 2.3. Expression Optimization of Recombinant Vectors

According to the results of the response surface analysis, the highest enzymatic activity of the EsFBA was observed at an induction temperature of 16 °C, an induction time of 16 h, and a final concentration of 0.55 mM of IPTG. The significance test analysis of the regression equation showed that the model was extremely significant (*p* < 0.001), and the missing items were not significant (*p* = 0.159 > 0.05). The data showed that the regression equation was a satisfactory fit, and the experimental error of the model was not significant; therefore, the regression equation can be used as a basis for predicting and analyzing the EsFBA enzyme activity [36]. The multiple quadratic regression model obtained was EsFBA activity = 1454.57 + 67.28A + 50.52B + 5.59C + 21.38AB − 23.96AC − 54.58BC − 352.09A2 − 249.28B2 − 268.25C2 (A: induction temperature; B: induction time; C: IPTG concentration). The F-values of the three influencing factors showed that the magnitude of the effect of each factor on the EsFBA enzyme activity was in the order A > B > C. All plots of the response surface were open downwards, with convex surfaces, and all had extreme values (Appendix A). The interaction of AB had a greater effect on the response value, the steepness of the entire surface was more pronounced, and the shape of the contours was close to elliptical, indicating that the interaction of the two factors was more significant.

### 2.4. Activity Analysis

The fructose-1,6-bisphosphate aldolase detection kit was used to determine the enzyme activity of the EsFBA. The purification multiple (specific activity after purification/specific activity before purification) and the recovery rate (total activity after purification/total activity before purification) of the purified enzyme were obtained (Table 1), that is, the EsFBA protein was purified 20.4-fold by HisTrap HP affinity chromatography. The specific activity of the purified enzyme was 193.5 U/mg.

The enzymatic activity of the EsFBA in this study was 18.1-fold higher than that of *Ascaris suum* (10.7 U/mg) [37], 15-fold higher than that of *Staphylococcus aureus* (12.9 U/mg) [38], 1.5-fold higher than that of blue-green algae [39], and 8.5-fold higher than that of sea turtles (22.8 U/mg) [19]; the high activity of the EsFBA may be related to the extremely low temperatures and long periods of diving time of *E. superba*.

### 2.5. Biochemical Characterization

The EsFBA enzyme activity increased as the reaction temperature increased from 10 °C to 45 °C, reaching a maximum activity at 45 °C (Figure 5a). The catalytic activity decreased while the temperature was greater than 45 °C. FBA from *E. superba* has a higher retained activity at higher temperatures compared to FBA from rabbit muscle (RAMA); EsFBA retained 20% activity at 70 °C, while RAMA lost almost all of its activity.

The pH of the reaction buffer is an important factor affecting the activity of the enzyme. The catalytic activity of the enzyme increased with an increase in pH between pH 4.0 and 7.0 (Figure 5b), reaching a maximum at pH 7.0. As the pH increased further between 7.0 and 9.0, the catalytic activity of the enzyme decreased. The conformation of the enzyme might change under different pH conditions, which could alter the binding or dissociation of the substrate from the active site of the enzyme, thereby affecting the enzyme activity.

The enzyme was highly stable at 20 °C and maintained 93.3% of its original activity after 2 h of incubation (Figure 5c). The enzyme activity was 70.6%, 65.4%, 7.1%, 1%, and 0.2% after 2 h of incubation at 30 °C, 40 °C, 50 °C, 60 °C, and 70 °C, respectively. In the pH stability test (Figure 5d), the enzyme activity could be maintained at 95% of its original activity after 2 h of incubation at pH 7.0. As acidity and alkalinity increased, the enzyme activity gradually decreased. At pH values of 4 and 9, the enzyme activity remained less than 70% after 2 h of incubation, indicating that EsFBA is a neutral enzyme.

It was demonstrated that the effect of 5 mmol/L of metal ions on the EsFBA enzyme activity was not significant. The relative enzyme activity after the influence of monovalent ions, such as K^+^ and Na^+^, was 125% and 122% (Figure 6). Meanwhile the divalent ions Cu^2+^, Mn^2+^, Mg^2+^, Ca^2+^, and Zn^2+^ had minimal effects on the enzyme activity: 111%, 107%, 96%, 120%, and 108%, respectively. More importantly, a 97% retention of the enzyme activity occurred under the influence of 5 mM EDTA, while 5 mM NaBH_4_ almost completely inhibited the EsFBA activity by 13%. The above results indicate that the addition of metal ions and EDTA did not significantly increase the enzymatic activity; on the contrary, the presence of borohydride significantly inhibited its enzymatic activity. It is possible that EsFBA does not necessitate the presence of metal ions as co-substrates, but rather catalyzes the reaction through Schiff base, indicating that EsFBA is a class I aldolase.

### 2.6. Site-Directed Mutagenesis

The D34K mutant exhibited catalytic ability (191.9 U/mg) similar to that of the wild type (193.5 U/mg) (Table 2). K147D showed 48% (93.2 U/mg) relative enzyme activity, E188H showed 74% (143.7 U/mg) relative enzyme activity, and K230D showed roughly half the activity (101.7 U/mg) of the wild type.

The evaluation of hydrogen bonding by PyMol showed that Lys147 and Lys230 are hydrogen-bonded to the substrate (Figure 7), which may be one of the reasons for the much-reduced enzyme activity after mutation at both sites. Furthermore, we deduced the active site of the substrate at the oxygen atom of its five-membered ring from the chemical equation of the FBP decomposition and measured the distance between the nitrogen atom of the mutant amino acid and the oxygen atom of the substrate action site (Figure 8). All mutated amino acids showed some degree of distance increase compared with the wild type, with D34K increasing from 4.9 to 6.4 Å, K147D increasing from 2.6 to 4.9 Å, E188H increasing from 3.1 to 4.6 Å, and K230D increasing from 4.9 to 5.9 Å. We speculate that the increase in the amino acid distance from the active site of the substrate also has an effect on enzyme activity. More importantly, they are involved in FBA binding to the substrate and Schiff base formation, which is perhaps the main reason for the reduced enzyme activity affecting the mutants. These results were consistent with previous studies [40,41,42]. Thus, based on results from the hydrogen-bond interaction analysis and changes in enzyme activity, Lys147 and Lys230 are the key amino acids of EsFBA.

## 3. Materials and Methods

### 3.1. Materials and Instruments

*E. superba* samples were kindly provided by the Liao Yu Group, who captured the samples in Antarctic waters in October 2019 and cryopreserved them at −80 °C until use. All chemicals were purchased from Sinopharm Reagent Co., Ltd. (Shanghai, China), Shanghai Sangon Biology Co., Ltd. (Shanghai, China), Dalian Takara Co., Ltd. (Dalian, China), and British OXOID Co., Ltd. (Basingstoke, UK) and used without further purification. Mettler Toledo S20 pH meter (Mettler Toledo, Greifensee, Switzerland), HH-1 electric constant temperature water bath (Changzhou Guohua, Changzhou, China), CR21GII high-speed refrigerated centrifuge (Hitachi, Tokyo, Japan), AKTA Explorer FPLC System (GE, Milwaukee, WI, USA), gel electrophoresis system (Bio-Rad, Hercules, CA, USA), and ultrapure water purification system (Millipore, Bedford, MA, USA) were used during the study.

### 3.2. Cloning and Bioinformatics Analysis

RNA was isolated from whole animals using the RNeasy system with on-column DNase treatment (Qiagen, Beijing, China). Genomic DNA was removed from total RNA by treatment with RNA-Free DNase (Thermo, Waltham, MA, USA). Reverse transcription reactions were performed using the PrimeScript 1st Strand cDNA Synthesis Kit (TaKaRa, kyoto, Japan). Then, the cDNA was used as a template to amplify the EsFBA gene, which was obtained through transcriptome sequencing. The EsFBA gene sequence has been submitted to NCBI, and the number is GenBank OP087496. Two primers, EsFBAF1(5′-GCGTCGCTAGCCACGC-3′) and EsFBAR1(5′-TTCCTGAGCCTTCCATC-3′), were synthesized. A polymerase chain reaction (PCR) was performed in a 25.0 μL reaction volume, containing 2.5 μL of 10 × PCR buffer, 2.5 μL of MgCl2, 2.0 μL of dNTP mix, 0.5 μL of Taq DNA polymerase (Qiagen), 15.5 μL of ddH2O, and 0.5 μL of each primer (10 mM). The PCR product was purified, subcloned into the pEasy-T vector (Qiagen), and transformed into the competent *E. coli* DH5α. The positive clone was sequenced to verify the full-length EsFBA cDNA. Based on the amino acid sequence, the molecular weight, isoelectric point, and affinity were predicted using ExPASy; the glycosylation site was predicted using NetNGlyc online; the 3D structure of the protein was predicted using SWISS-MODEL; and the BLAST comparison was performed using NCBI. The amino acid sequences of translated EsFBA were compared with DNAMAN, homology analysis was performed, and a phylogenetic tree was constructed using MEGA. Molecular docking simulations of EsFBA and substrate FBP were performed using Autodock software.

### 3.3. Construction of Recombinant Plasmid

We constructed a C-terminal 6× His-tagged sfGFP-EsFBA fusion protein expression plasmid using *E. coli* pET-28a; sfGFP has high solubility, bright fluorescence, fast-folding ability, high resistance to denaturants, and can be used as a fusion tag to promote the folding of heterologous proteins in *E. coli* [43,44]. The DNA fragments encoding the EsFBA tag and HRV 3C recognition site were amplified from the plasmid EsFBA by PCR using the primers 3C-EsFBA-F and EsFBA-R (Appendix A). The DNA fragments, including the HRV 3C recognition site and sfGFP, were amplified from the plasmid sfGFP by PCR using the primers sfGFP-F and sfGFP-3C-R. The DNA fragment encoding sfGFP-3C-EsFBA was amplified from the above EsFBA and sfGFP PCR products using primers EsFBA-R/sfGFP-F and cloned into pET28a-T vector employing Nco I and Xho I. The plasmid was named pET28a-sfGFP-EsFBA.

### 3.4. Protein Expression and Purification

The plasmid pET28a-sfGFP-EsFBA was transformed into BL21(DE3) competent cells and transfected on Luria Bertani (LB) broth plates containing 100 µg/mL kanamycin. Individual colonies of BL21(DE3) cells were then cultured overnight on plates in a 250 mL flask containing 50 mL LB broth (containing 100 µg/mL kanamycin) at 37 °C and 200 rpm. Precultured cells were inoculated in 1 L flasks containing 100 mL LB and 100 μg/mL kanamycin and were grown at 37 °C and 200 rpm. Protein expression was induced by the addition of IPTG at a final concentration of 0.2 mM for 18 h at 16 °C. Cells were incubated at 200 rpm and 16 °C for 18 h. Cells were collected via centrifugation at 7654× *g* for 15 min, and the cell pellet was collected in PBS (135 mM NaCl, 2.7 mM KCl, 1.5 mM KH_2_PO_4_, and 8 mM K_2_HPO_4_; pH 8.0).

The cultures were washed twice in PBS and resuspended in 25 mL of PBS buffer containing 1 mM PMSF and 10 mM imidazole. The solution was sonicated for 30 min and centrifuged at 14,000 rpm for 0.5 h at 4 °C. The supernatant was subjected to HisTrap HP affinity chromatography, and the target protein was separated from the supernatant. The final purification was performed on the AKTA fast liquid chromatography system (GE Healthcare) using a PBS loading buffer and PBS elution buffer containing 500 mM imidazole at a flow rate of 1.0 mL/min. The first peak appeared as a penetration peak, followed by a gradual gradient elution to collect the target protein. The obtained samples were ultra-filtered at 30 kD, 4000 rpm, and imidazole was removed by the successive addition of PBS. The samples were analyzed for purity using 10% SDS-PAGE.

### 3.5. Optimization of Induction Conditions

The induction conditions were optimized using a response surface analysis and a three-factor three-level experimental design was carried out using Design Expert 10.0. Three aspects of optimal conditions were explored, including induction–inducer concentration (0.1–1 mmol/L), induction duration (12–20 h), and induction temperature (12–20 °C). Induction conditions for the highest enzyme activity were selected.

### 3.6. Activity Assay

The protein concentration was determined using the BCA protein assay kit (Solarbio, Beijing, China), and OD562 was monitored to calculate protein concentration. The enzyme activity was detected by fructose-1,6-bisphosphate aldolase assay kit (Figure 9) (Solarbio). FBA catalyzes the formation of FBP to glyceraldehyde 3-phosphate and dihydroxyacetone phosphate. Under the action of propanose phosphate isomerase and α-glycerol phosphate dehydrogenase, NADH and dihydroxyacetone phosphate are catalyzed to generate NAD+ and glycerol 3-phosphate. Since NADH has absorbance at 340 nm, the change in absorbance value at 340 nm is the change of NADH into NAD+. Subsequently, the activity of FBA can be reflected. Consumption of 1 nmol of NADH per mg of protein per minute was defined as one unit of enzyme activity. Briefly, 900 μL reagent and 100 μL protein samples were added to a colorimeter; absorbance was detected within 5 min at 340 nm and 25 °C. Three parallel groups were set up for each experiment.

### 3.7. Biochemical Characterization

Purified EsFBA was incubated at different temperatures (10 °C, 20 °C, 30 °C, 40 °C, 45 °C, 50 °C, 60 °C, and 70 °C) for 5 min at pH 7, followed by an activity assay using FBP as a substrate to determine the optimal temperature. The enzyme solution was placed in buffer solutions at different pH values (4.0, 5.0, 6.0, 7.0, 8.0, and 9.0) to determine the optimal pH, with 50 mmol/L Na_2_HPO_4_ citric acid buffer (pH 5.0–8.0) and 50 mmol/L glycine–NaOH buffer (pH 9.0, 50 mM). Purified EsFBA was placed in a metal bath at each temperature. The change in enzyme activity was measured at each temperature over 120 min; the highest enzyme activity was taken as 100%. Purified EsFBA was placed in a metal bath at 25 °C, and the enzyme activity at each pH was measured over 120 min, with the highest enzyme activity at 100%, as a test of the pH stability of EsFBA.

The effect of metal ions (Ca^2+^, K^+^, Na^+^, Mn^2+^, Cu^2+^, Mg^2+^, and Zn^2+^), EDTA, and NaBH_4_ with the final concentration of 5 mM was determined at pH 7.0 and 25 °C. The above solution was mixed with purified EsFBA in the appropriate proportions and left to stand for 2 h at 25 °C. The residual activity of EsFBA was assayed at 25 °C and pH 7. EsFBA, without the addition of metal ions and chemical reagents, was used as a control, and its activity was considered 100%.

### 3.8. Site-Directed Mutagenesis

Numerous studies have shown that amino acid residues inside neutral sites are highly conserved throughout evolution; Glu187, a conserved amino acid located in the active site of fructose-1,6-bisphosphate aldolases [45], plays a variety of roles in the catalytic cycle of aldolases. The residue responsible for the nucleophilic attack of the ketone substrate during Schiff base formation is located inside the active site of Lys229 [41]; Asp33 is involved in the extraction of protons from the C4 hydroxyl group [46], and studies suggest that this residue plays a key role in the catalytic mechanism. Lys146 is absolutely conserved in class I aldolases and is associated with the catalytic role of protein modifications [47]. Mutation of the above amino acid residues could verify the key amino acid sites of EsFBA and also confirm EsFBA’s relationship with other class I FBAs. Thus, we prepared four single mutants (D34K, K147D, E188H, and K230D) and investigated the catalytic ability of the enzyme.

### 3.9. Molecular Modeling

The tertiary structure of EsFBA was built by I-TASSER, which uses a hierarchical approach to protein structure prediction. The predicted structure was constructed using FBA (PDB code: 1FBA) as a template. The best model was further assessed using the PROCHECK plot in order to analyze the quality and consistency of the generated model. The structures of the D34K, K147D, E188H, and K230D mutants were simulated based on the predicted crystal structure of FBA using Insight II 2000/Discover 3 with Extensible Systematic Force Field. We obtained a unique structure for each mutant starting from several initial conformations. These structures were employed as rigid receptors for substrate docking. The Autodock 4 program was used to perform the docking of FBP into the active site of FBA mutants. The amino acid residues Lys108, Lys230, Gly302, and Ala304 were set as flexible residues. The ligand FBP for docking was generated by Dundee PRODRG server. Docked conformations were analyzed and ranked automatically by Autodock 4 using a free-energy scoring function. These results were then visualized using PyMol. Then, we used the Q5 site-directed mutagenesis kit, which contains 12.5 µL of master mix, 1 µL 10 µM of forward primer, 1 µL 10 µM of reverse primer, 1 µL of template DNA, and 9 µL of nuclease-free water. After PCR and KLD reaction, we performed receptor-cell ligation and transformation to obtain the mutant strain.

## 4. Conclusions

In this study, the FBA gene from *E. superba* was cloned. The full-length cDNA sequence of EsFBA is 1098 bp, encoding a protein of 365 amino acids. By comparing multiple sequences and building a phylogenetic tree, we found that the EsFBA had the highest homology (84.38%) with the FBA of *P. vannamei*. We expressed the EsFBA in *E. coli* and obtained pure protein. The optimum temperature of the recombinant EsFBA was 45 °C, and the optimum pH was 7. While 5 mM of metal ions and EDTA had little effect on the EsFBA enzymatic activity, the borohydride inhibited it, demonstrating that EsFBA is a class I aldolase. Furthermore, point mutations allowed us to verify important sites of EsFBA activity, with mutants showing a reduced catalytic activity compared to the wild type. Compared with aldolase from rabbit muscle, EsFBA retained relatively high activity at a high temperature, and, compared with other species, the activity of the FBA from *E. superba* was more than ten times higher. Therefore, EsFBA could be used as a potential new enzyme for development and utilization. This study provides a theoretical basis for the use of *E. superba*.

## Figures and Tables

**Figure 1 ijms-23-10478-f001:**
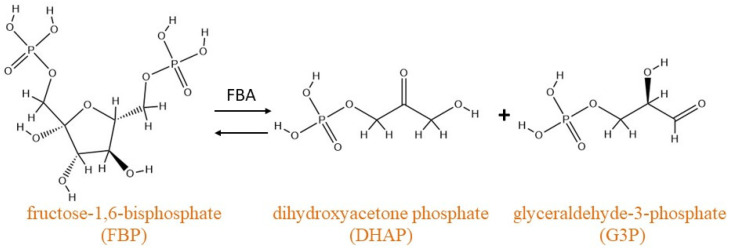
Chemical catalyzed by fructose-1,6-bisphosphate aldolase (FBA). FBA catalyzes the cleavage of fructose 1,6-diphosphate (FBP) to dihydroxyacetone phosphate (DHAP) and glycerol 3-phosphate (G3P).

**Figure 2 ijms-23-10478-f002:**
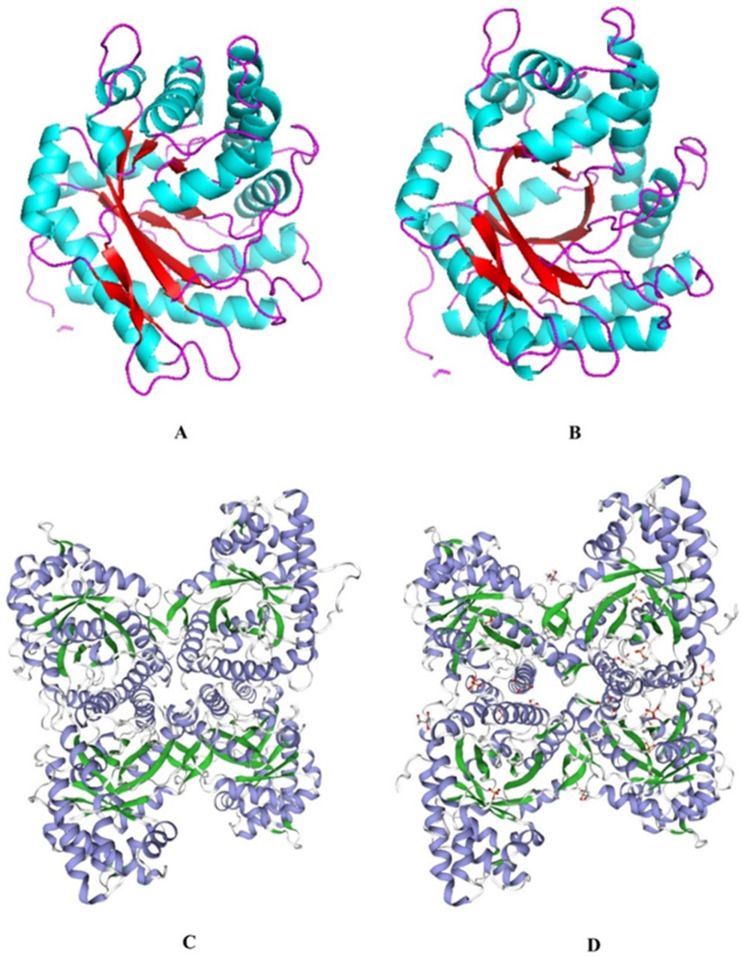
EsFBA 3D structure prediction diagram. (**A**) 3D structure prediction map of *Euphausia superba* FBA; (**B**) human A-type FBA 3D structure template; (**C**) 3D structure prediction map of the FBA tetramer in *Euphausia superba*; (**D**) template diagram of the human FBA-A 3D structure.

**Figure 3 ijms-23-10478-f003:**
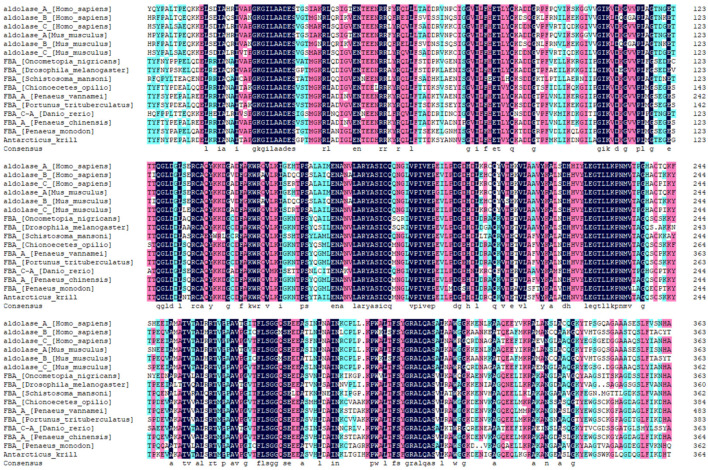
Multiple-sequence alignment of EsFBA and other reported fructose-1,6-bisphosphate aldolases. Conserved amino acids are highlighted in black.

**Figure 4 ijms-23-10478-f004:**
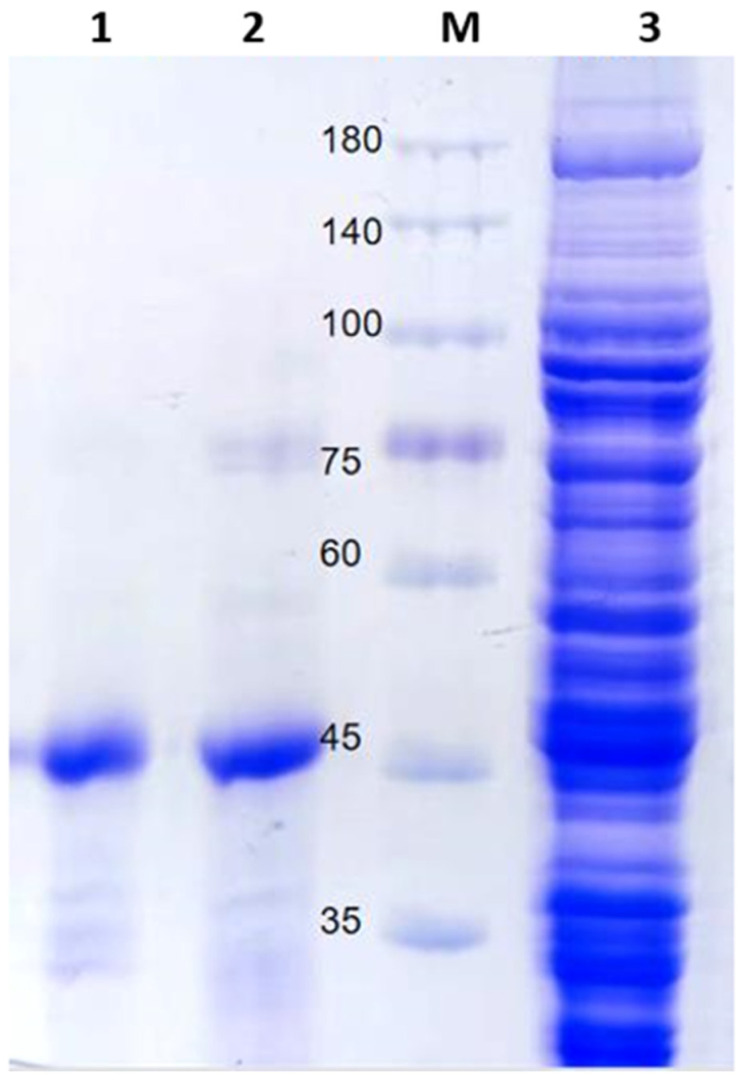
Sodium dodecyl sulfate–polyacrylamide gel electrophoresis analyses of recombinant EsFBA on a 10% gel. M: marker; Lane 1: protein purified from 500 mM imidazole; Lane 2: protein purified from 500 mM imidazole; Lane 3: supernatant.

**Figure 5 ijms-23-10478-f005:**
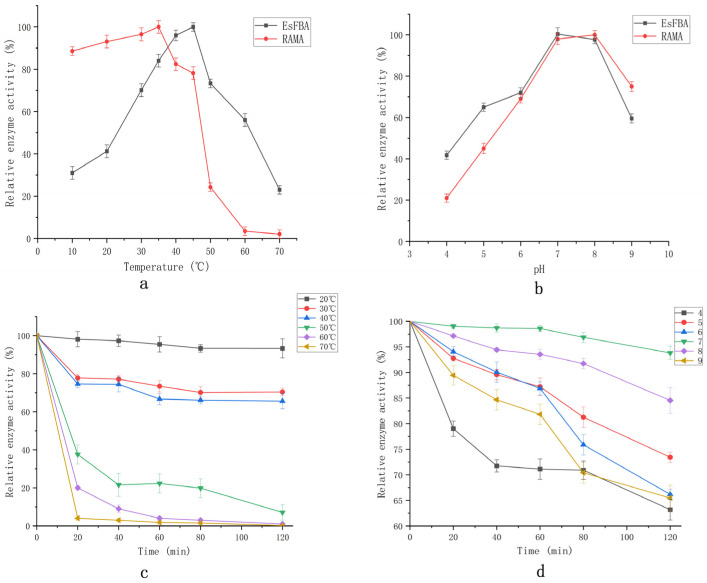
Characterization of recombinant EsFBA. (**a**) Optimum temperature, (**b**) optimum pH, (**c**) temperature stability, and (**d**) pH stability.

**Figure 6 ijms-23-10478-f006:**
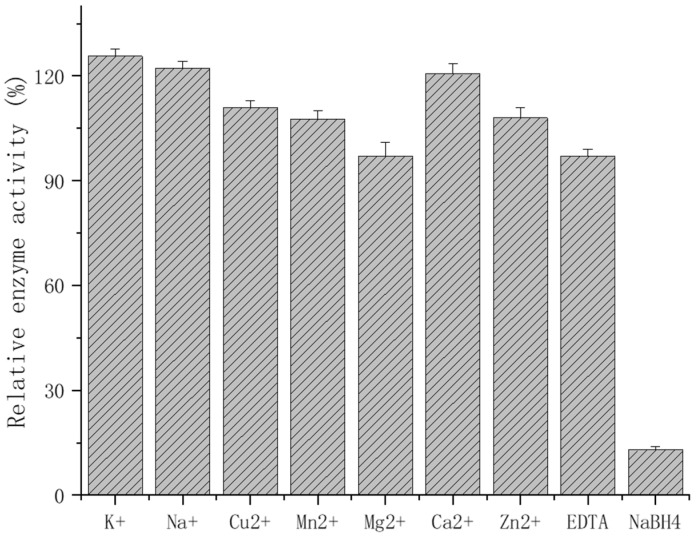
EsFBA activity in the presence of metal ions.

**Figure 7 ijms-23-10478-f007:**
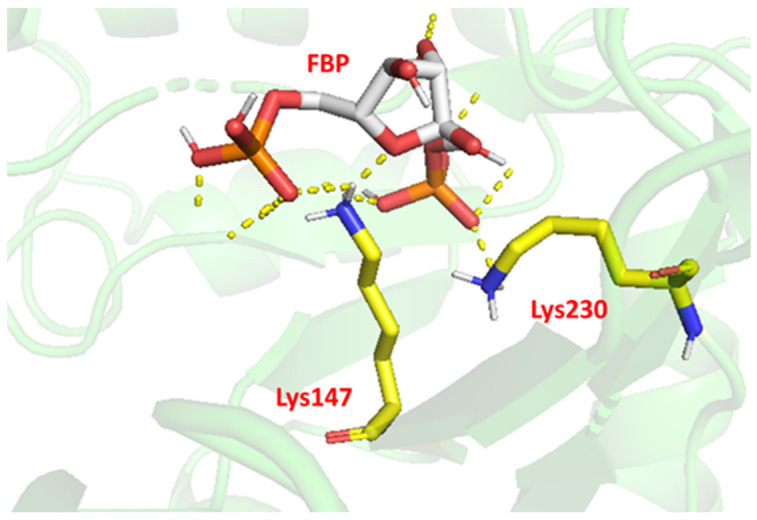
Substrate hydrogen-bonding simulation diagram. Yellow dashed lines represent the hydrogen bonds interacting with the substrate fructose-1,6-bisphosphate (FBP), where Lys147 and Lys230 are hydrogen-bonded to the substrate.

**Figure 8 ijms-23-10478-f008:**
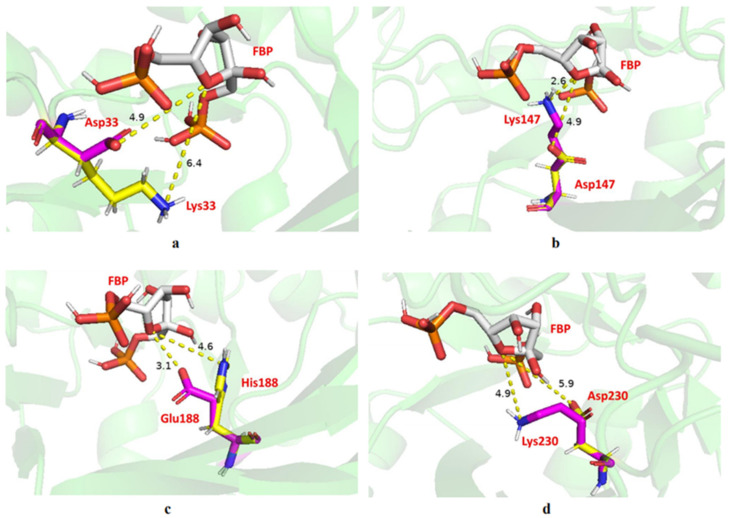
Diagram showing the interaction between mutated or wild-type amino acids with the substrate. (**a**) D34K, (**b**) K147D, (**c**) E188H, and (**d**) K230D.

**Figure 9 ijms-23-10478-f009:**
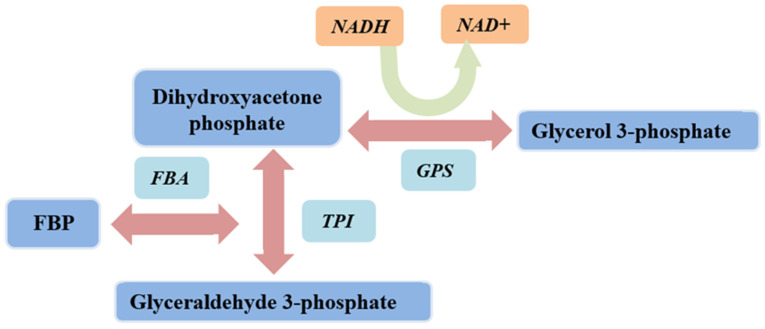
Catalytic processes involving fructose-1,6-bisphosphate aldolase. FBA reversibly catalyzes the cleavage of FBP to DHAP and G3P, where G3P is converted to DHAP via TPIase and DHAP is converted to glycerol 3-phosphate by GPS. NADH is converted to NAD+ in the process.

**Table 1 ijms-23-10478-t001:** Purification of recombinant EsFBA.

	Total Activity (U)	Protein Concentration (mg/mL)	Specific Activity (U/mg)	Purification Fold	Yield (%)
Crude enzyme	799.8	8.1	9.5	1	100
Ni^2+^-NTA	280.4	0.7	193.5	20.4	35.1

**Table 2 ijms-23-10478-t002:** Enzyme activity of wild type and mutant.

	Enzyme Activity (U/mg)	Relative Enzyme Activity
Wild type	193.5 ± 0.37	100%
D34K	191.9 ± 1.05	99%
K147D	93.2 ± 1.34	48%
E188H	143.7 ± 0.98	74%
K230D	101.7 ± 0.72	53%

## Data Availability

Not applicable.

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
