# Peer review of "Cloning and Characterization of Fructose-1,6-Bisphosphate Aldolase from Euphausia superba"

_ijms, 2022, doi:10.3390/ijms231810478_

Round 1

Reviewer 1 Report

The authors cloned and characterized a very interesting enzyme that is similar in structure to class I fructose-1,6-bisphosphate aldolase from the small marine planktonic crayfish Euphausia superba, but it may be a metal-dependent enzyme as representative of class II enzymes.

Major points

So, can EsFBA be classified as a class II enzyme? Do class II enzymes have any specific sequence features? Does the EsFBA sequence have any such features?

The authors need to describe in detail the rationale for choosing amino residues for mutagenesis.

Since enzyme activity is highly dependent on metals, the authors need to conduct an experiment with EDTA.

The authors need to be clearer in their conclusions about the unusual properties of the enzyme they characterized.

Minor Comments.

The authors need to mention Class IA FBA more clearly in the introduction (as a source example, e.g., 10.3389/fmolb.2021.719678).

Line 51 There is a typo in the word "mulecules".

Line 69 Please correct the phrase "has properties of optimal temperature at low temperature".

Line 71 Better say E.superba autoproteolysis rather than degradation.

Lines 88-89. Please state the properties of the protein more correctly: does it have hydrophobic or hydrophilic domains? Or does it have a hydrophilic core and a hydrophilic surface?

Lines 102,103. Species names should be italicized.

Table 1 Do the authors mean "specific activity" instead of "specificity activity"?

Line 264 Do the authors mean "chemocompetent" instead of the strange term "chemosensory"?

Line 294 Do the authors need to use a period instead of a comma to separate sentences?

Figure 9 Depicts a biochemical process, but the authors need to show the scheme of the fructose-1,6-bisphosphate aldolase assay kit, specifically how the signal is generated.

The References section needs revision: there are a number of [J] signs, several authors' names and the names of journals are completely capitalized.

Reviewer 2 Report

This paper describes the characterization of the fructose-1,6-bisphosphate aldolase from E. superba. It is an interesting paper that presents novel and important findings; it is concise and well written. However, the data presented is not robust as it is, and I have a few suggestions to improve the manuscript.

-Results:

Figures 5 and 6 – Were these experiments run in replicates? I would strongly suggest running these experiments in technical triplicates to check the variability of the assay, especially because this is the first report on this enzyme. I wouldn’t trust the data if there’s no reproducibility.

Additionally, it would add robustness if a positive control were added to the assays – for example, a well-studied aldolase with known and reproducible activity. This way I would believe your assay is working properly and the numbers are real.  

Also, were the mutant activities repeated or run in replicates? (table 2). I also would like to see the variability in this assay. The methods say that three parallel groups were done for each experiment; if that is the case, the results should be added to the plots, preferably as single values and with a deviation value.  

The authors could compare and discuss the optimal enzyme activity conditions obtained for EsFBA with FBA from other species – is the optimal temperature indeed lower than from other species, considering that E. superba comes from a cold environment? How is the stability compared to others?

Line 153 – It could be interesting to discuss the activity of FBA from other species besides turtles.

I suggest moving the Conclusions section after the results/conclusion and leaving the methods for last or after the introduction. The way the manuscript is presented induces the reader to think the paper has no conclusions.  

-Methods:

Site-directed mutagenesis was not described in the methods section. Please explain the techniques used.  

There is no statistical analysis described in the methods section. Were there no statistical analyses in the manuscript?

Please review all species names throughout the manuscript; they should be in italics.

Line 121 – “As described above “; probably should change to “as described in the methods section”.

Round 2

Reviewer 1 Report

The authors have substantially corrected the manuscript in accordance with the previous comments. I now have no further comments and can recommend the revised manuscript for publication.

Author Response

Thank you very much for your previous valuable comments.

Reviewer 2 Report

The authors have addressed all my concerns. I only suggest to review for typos (line 420, "ues"). Also, authors should correct “specificity activity” to “specific activity” as suggested by reviewer 1 (lines 163 and 164).

Author Response

As the reviewer suggested, we have changed "ues" to "used" in line 420, and we also corrected “specificity activity” to “specific activity” in line 163 and 164.